Taxonomy of Platypterygius campylodon and the diversity of the last ichthyosaurs

Fischer Valentin v.fischer@ulg.ac.be
UR Geology, Université de Liège , Liège , Belgium
Farke Andrew
Electronic publication date: 2016 Oct 20
Publication date: 2016
Volume: 4
Electronic Location ID: e2604
Received 2016 Aug 12; Accepted 2016 Sep 24
Copyright: ©2016 Fischer
Copyright year: 2016
Copyright holder: Fischer
License: This is an open access article distributed under the terms of the Creative Commons Attribution License, which permits unrestricted use, distribution, reproduction and adaptation in any medium and for any purpose provided that it is properly attributed. For attribution, the original author(s), title, publication source (PeerJ) and either DOI or URL of the article must be cited.
License URL: https://creativecommons.org/licenses/by/4.0/

Keywords: Ichthyosauria, Cretaceous, Extinction, Ophthalmosauridae, Platypterygiinae, Biodiversity, Marine reptiles, Feeding ecology

Funding: F.R.S.–FNRS (Belgium) Royal Society UK NF140022 I received support from an Aspirant and a Chargé de Recherches grant from the F.R.S.–FNRS (Belgium) and a Newton International Fellowship grant from the Royal Society (UK; NF140022) during this research. The funders had no role in study design, data collection and analysis, decision to publish, or preparation of the manuscript.

==============================
A complex and confusing taxonomy has concealed the diversity dynamics of Cretaceous ichthyosaurs (Reptilia) for decades. The near totality of Albian-Cenomanian remains from Eurasia has been assigned, by default, to the loosely defined entity Platypterygius campylodon, whose holotype was supposed to be lost. By thoroughly examining the Cenomanian ichthyosaur collections from the UK, I redescribe the syntypic series of Platypterygius campylodon. This material, along with a handful of other coeval remains, is diagnostic and seemingly differs from the vast majority of Cretaceous remains previously assigned to this taxon. A lectotype for Platypterygius campylodon is designated and I reassign this species to Pervushovisaurus campylodon nov. comb. The feeding ecology of this species is assessed and conforms to the scenario of an early Cenomanian diversity drop prior to the latest Cenomanian final extinction.

Introduction

Ichthyosaurs are iconic reptiles of the Mesozoic marine ecosystems that disappeared quite abruptly at the beginning of the Late Cretaceous (Bardet, 1992; Fischer et al., 2016). Understanding of the final chapter of their extensive evolutionary history (Olenekian-Cenomanian, about 157 million years (Bardet, 1992; Motani et al., 2015)) has been impaired by a complex and confusing taxonomy, especially at the supra-specific level. The genus Platypterygius is among the most problematic, with no robust phylogenetic definition, no diagnostic features and a biozone spanning the Barremian (‘Platypterygius’ sachicarum) to the Late Cenomanian (‘Platypterygius’ campylodon, ‘Platypterygius kiprijanoffi’), i.e., 35 million years (Fischer, 2012; Fischer et al., 2014a). Many recent phylogenetic analyses have found the species currently referred to Platypterygius to be widely scattered, sometimes within a particular ophthalmosaurid subfamily, Platypterygiinae (Druckenmiller & Maxwell, 2010; Fischer et al., 2012; Fischer et al., 2016; Zverkov et al., 2015). The type species of the genus, Platypterygius platydactylus, is phylogenetically isolated from other species currently referred to as Platypterygius (Fischer et al., 2016). Moreover, the taxonomy of Platypterygius might be biased by ecological convergence (Fischer et al., 2016). Thus, the diversity dynamics of Cretaceous ichthyosaurs cannot be approximated using currently valid genera; the taxonomy of each species needs to critically assessed in isolation and the use of the genus Platypterygius should be motivated with respect to the morphology of the type species. Other genus-group names have been used for mid Cretaceous ichthyosaurs in the past but have since been discarded, notably Myopterygius Huene, 1922, Tenuirostria (Arkhangelsky, 1998) and Longirostria (Arkhangelsky, 1998) (Huene, 1922; Arkhangelsky, 1998), adding to the confusion.

A persisting issue in quantifying the diversity and extinction tempo of the last ichthyosaurs is Ichthyosaurus campylodon Carter, 1846a, which has been used since its creation (Carter, 1846a; Carter, 1846b) as a bin for nearly all Cretaceous ichthyosaur remains from Eurasia, regardless of their morphology or stratigraphic position. In this brief contribution, I: (i) review the status and morphology of the syntypic material of Ichthyosaurus campylodon and other remains from the Cenomanian deposits of the United Kingdom; (ii) rediagnose and designate a lectotype for I. campylodon, transfer it to Pervushovisaurus campylodon nov. comb and discuss the status and availability of the genus-group taxon Myopterygius Huene, 1922; and (iii) assess the ecological diversity of the last ichthyosaurs by the means of a cluster dendrogram analysis of ecomorphological data.

Material and Methods

Specimen list–I surveyed the entire Cenomanian collections of the CAMSM, the RBINS and the NHMUK, but only important specimens are listed here (Table 1). Unlisted remains include centra, undeterminable skeletal fragments and poorly preserved isolated teeth. Specimens from Cambridge Greensand Member (i.e., the base of the West Melbury Marly Chalk Formation, Grey Chalk Subgroup (Hopson, 2005)) have been published elsewhere (Fischer et al., 2012; Fischer et al., 2014b) and are not listed here. I also briefly re-assess the morphology of the specimen used by Broili (1908) to erect the species Ichthyosaurus kokeni from the Hauterivian of Germany. This species has been regarded by Huene (1922) as belonging to Myopterygius, so it is relevant to discuss its status and morphology here.

Table 1 Important West Melbury Marly Chalk Formation specimens studied here.

Specimen	Material	Assignation	Locality	
CAMSM B20643	Tooth	Platypterygiinae indet. (holotype of I. angustidens= nomina nuda Fischer et al. (2014b))	Hunstanton	
CAMSM B20644	Tooth	Pervushovisaurus campylodon (syntype, Carter’s series)	Cambridge area	
CAMSM B20645	Tooth	Platypterygiinae indet. (syntype, Carter’s series)	Cambridge area	
CAMSM B20646	Tooth	Pervushovisaurus campylodon (syntype, Carter’s series)	Cambridge area	
CAMSM B20647	Tooth	Pervushovisaurus campylodon (syntype, Carter’s series)	Cambridge area	
CAMSM B20648	Tooth	Pervushovisaurus campylodon (syntype, Carter’s series)	Cambridge area	
CAMSM B20649	Tooth	Pervushovisaurus campylodon (syntype, Carter’s series)	Cambridge area	
CAMSM B20650	Tooth	Pervushovisaurus campylodon (syntype, Carter’s series)	Cambridge area	
CAMSM B20651	Tooth	Pervushovisaurus campylodon (syntype, Carter’s series)	Cambridge area	
CAMSM B20652	Tooth	Pervushovisaurus campylodon (syntype, Carter’s series)	Cambridge area	
CAMSM B20653	Tooth	Pervushovisaurus campylodon (syntype, Carter’s series)	Cambridge area	
CAMSM B20654	Tooth	Pervushovisaurus campylodon (syntype, Carter’s series)	Cambridge area	
CAMSM B20655	Tooth	Pervushovisaurus campylodon (syntype, Carter’s series)	Cambridge area	
CAMSM B20656	Tooth	Pervushovisaurus campylodon (syntype, Carter’s series)	Cambridge area	
CAMSM B20657	Tooth	Pervushovisaurus campylodon (syntype, Carter’s series)	Cambridge area	
CAMSM B20658	Tooth	Pervushovisaurus campylodon (syntype, Carter’s series)	Cambridge area	
CAMSM B20659	Partial rostrum	Pervushovisaurus campylodon (syntype, Carter’s series)	Cambridge area	
CAMSM B20671a	Rostrum	Pervushovisaurus campylodon	Barrington	
CAMSM B75736	Atlas-axis	Ichthyosauria indet.	Cambridge area	
CAMSM B42257	Centrum	Ichthyosauria indet.	Hunstanton	
CAMSM TN282	Rostrum	Pervushovisaurus campylodon	(chalky part of the Cambridge Greensand Member, Cambridge area)	
CAMSM TN283	Rostrum	Platypterygiinae indet.	(chalky part of the Cambridge Greensand Member, Cambridge area)	
CAMSM unnumbered	Humerus (HM1 morphotype of Fischer et al. (2014b))	Platypterygiinae indet.	Cambridge area	
NHMUK 5648	Teeth	Platypterygiinae indet.	?	
NHMUK 33294 partim	Teeth	Platypterygiinae indet.	Isleham, Cambridgeshire	
NHMUK 41367	Anterior tip of rostrum	Platypterygiinae indet.	?	
NHMUK 41895	Anterior tip of rostrum	Platypterygiinae indet.	?	
NHMUK R13	Teeth	Platypterygiinae indet.	?	
NHMUK R49	Teeth	Platypterygiinae indet.	Lyden Spout, Folkestone	
NHMUK R2335	Rostrum	Platypterygiinae indet.	?	
NHMUK R2385	Fragmentary rostrum	Platypterygiinae indet.	?	

Late Cretaceous ichthyosaur feeding guilds–I updated the ecological dataset of Fischer et al. (2016) to assess the feeding guilds and ecological diversity of the last ichthyosaurs. This dataset is composed of a series of ecologically-relevant measurements and ratios: absolute tooth size, crown shape (height/basal diameter), crown height relative to basioccipital diameter, relative symphysis length, relative snout depth, absolute aperture of the sclerotic ring and quantification of tooth wear. I have added novel data on the symphysis of ‘Platypterygius’ sachicarum (E Maxwell pers. comm., 2016) corrected an erroneous value on the symphysis of the ‘Platypterygius’ hercynicus and added Pervushovisaurus campylodon to the dataset. The details of the specimens used and explanation for each character is given in Supplemental Information 1; the dataset itself is available as Data S1. As in the original publication, I submitted this data set to a cluster dendrogram analysis in R using the Ward method. Data were scaled to have equal variances and transformed to a Euclidean distance matrix before clustering. Because the data is restricted to ecologically relevant measurements and with a strong emphasis on Cretaceous forms, the resulting dataset is small and contain a non-negligible proportion of missing values (39%), which renders usual bootstrapping methods inadequate. To cope with this issue, I assessed the statistical support of our cluster using the “Approximately Unbiased P-value” method of the pvclust v2.0–0 package (Suzuki & Shimodaira, 2015) in R. This method employs multiscaled bootstrapping: instead of simply bootstrapping the dataset, it creates multiple datasets that are smaller, equal and larger than the original dataset. I ran it from 0.5 times to 5 times the size of the original dataset, with 0.1 increments and 10,000 bootstrap per increment.

Nomenclatural acts–The electronic version of this article in Portable Document Format (PDF) will represent a published work according to the International Commission on Zoological Nomenclature (ICZN), and hence the new names contained in the electronic version are effectively published under that Code from the electronic edition alone. This published work has been registered in ZooBank. Zoobank does not currently support nomenclatural acts that do not establish new taxa (e.g., nov. comb.), so the specific nomenclatural acts of this paper cannot be entered in Zoobank for the time being. The ZooBank LSIDs (Life Science Identifiers) can be resolved and the associated information viewed through any standard web browser by appending the LSID to the prefix “http://zoobank.org/”. The LSID for this publication is: urn:lsid:zoobank.org:pub:019DACEA-EBBE-4FAE-B885-3A9E5B1E1315. The online version of this work is archived and available from the following digital repositories: PeerJ, PubMed Central and CLOCKSS.

History and Status of Platypterygius Campylodon

Carter (1846a) established the name Ichthyosaurus campylodon in a conference abstract. His initial description is based on an articulated rostrum with numerous teeth that he described in a paper the same year (Carter, 1846b). In that paper, he figured two teeth and made clear that his collection contained several specimens, coming from both the base (Cambridge Greensand Member) and the rest of the West Melbury Marly Chalk Formation. The Cambridge Greensand member mixes earliest Cenomanian specimens with reworked fossils from the Late Albian of the underlying Gault Formation (Hopson, 2005; Fischer et al., 2014b). It is therefore difficult to know which particular specimen was used to establish the species in Carter’s conference abstract, but relevant information can be extracted from the specimens from his collection, which are now housed in the Sedgwick Museum of the University of Cambridge, UK (CAMSM).

Huene (1922) assigned Ichthyosaurus campylodon to the genus Myopterygius Huene, 1922 and created another genus, Platypterygius Huene, 1922 for reception of a single species from the Lower Aptian of Germany, Platypterygius platydactylus (Broili, 1907). McGowan (1972) then transferred all species belonging to Myopterygius to Platypterygius. He choose Platypterygius over Myopterygius as the single valid Cretaceous ichthyosaur genus “Because platydactylus is the best known species, the genus Platypterygius is the most appropriate” (McGowan, 1972: 18). Since Carter’s and McGowan’s publications, an overwhelming amount of Cretaceous ichthyosaur remain from Eurasia has been referred to Platypterygius campylodon, mostly by default (e.g., Kiprijanoff, 1881; Kiprijanoff, 1883; Sauvage, 1882; Delair, 1960; Buffetaut, 1977; Buffetaut et al., 1981; Buffetaut, Tomasson & Tong, 2003). Some remains were referred to the species Platypterygius kiprijanoffi (Romer, 1968; Bardet, 1989), but these were subsequently assigned to as Platypterygius campylodon by McGowan & Motani (2003).

Currently, Platypterygius campylodon is a vague entity with no clear-cut morphology nor any valid diagnostic feature, itself included in a poorly defined genus. As a matter of fact, the only diagnostic feature proposed by McGowan & Motani (2003) for Platypterygius campylodon is the probable presence of an “External longitudinal groove […] along the length of the rostrum and mandible” (=fossa praemaxillaris/dentalis); such sulcus is actually present in all neoichthyosaurians I have examined so far. With no holotypic or syntypic material clearly identified as such and no diagnostic feature, this species had to be considered as a nomen dubium.

Figure 1 Syntypic material of Pervushovisaurus campylodon (Carter, 1846a).

(A) CAMSM B20645 a posterior tooth likely to be the one figured by Carter (1846b), which is reproduced in (B). This tooth cannot be unambiguously referred to I. campylodon and is regarded as Platypterygiinae indet. (C) CAMSM B20644, a large mid-snout tooth, likely to be the one figured by Carter (1846b), which is reproduced in (D). (E–G) Teeth from CAMSM B20659, a partial rostrum; this specimen was figured by Owen (1851) (Pl. XXV) and is here selected as the lectotype. (E) Small posterior tooth from CAMSM B20659. (F) Mid-snout dentary teeth from CAMSM B20659. White arrows indicate curved tooth roots in the lower jaw, considered by Carter (1846b) (and subsequent authors) as a diagnostic feature. This feature is here regarded as doubtful and appears to be diagenetic. (G) Mid-snout premaxillary teeth from CAMSM B20659. (H) Small mid-snout tooth (CAMSM B20646) illustrating the sharp angle ridges on the root.

McGowan & Motani (2003) attempted to solve this issue. They regarded the specimen SMC B20644 (=CAMSM B20644), “a 60-cm rostral fragment”, as the presumed holotype for Ichthyosaurus campylodon, mainly because its size matched the length given by Carter (“more than 2 feet”, p7 in Carter, 1846b). But there are several issues with that decision. Firstly, CAMSM B20644 is not a 2 feet-long rostrum but an isolated tooth from Carter’s Ichthyosaurus campylodon collection; this tooth actually seems to be the tooth figured by Carter (Figure a in Carter (1846b: 6); see Fig. 1), as already suggested by Delair (1960). McGowan & Motani (2003: 120) actually figured a portion of another specimen, CAMSM B20671. CAMSM B20671 is actually more complete than figured in McGowan & Motani (2003) and has diagnostic features (see below), but that specimen is 790 mm, i.e., 2.59 feet long. CAMSM B20671 preserves the tip of both the rostrum and the mandible, whereas Carter clearly stated that the specimen he described lacked these parts (Carter, 1846b: 7). Moreover, CAMSM B20671 is from Barrington quarry and the date written on the specimen is 1881, 35 years after Carter’s original descriptions. While this date may be the acquisition date by the museum, all specimens from Carter’s collection have a green label glued on them containing “Presented by J. Carter Fsq.ES.G” and CAMSM B20671 lacks such a label. Actually, there is not a single 2 feet long rostrum in the CAMSM that bears such label. Delair (1960) listed B.20644-59 C.U.M. B.58379-82 C.U.M (CAMSM B20644_59 and CAMSM B58379_82) as “types” of Ichthyosaurus campylodon, but without any justification or argument for this a posteriori designation. CAMSM B58379_82 are phosphatized teeth from the Cambridge Greensand Member and are therefore reworked from the upper part of the Gault formation (Upper Albian, see Fischer et al. (2014b) for a treatment of these remains), while CAMSM B20644_59 are teeth from the unreworked part of the West Melbury Marly Chalk Formation and are Early Cenomanian in age (Hopson, 2005). This, again, contradicts Carter’s account (1846a), which specifically discussed the morphology of a partial rostrum with associated teeth.

Figure 2 Designated lectotype for Pervushovisaurus campylodon (Carter, 1846a), CAMSM B20659.

(A) Mid-snout fragment in right lateral view, showing the diagenetically deformed dentary teeth. (B) Same fragment in dorsolateral view.

There are other large rostra lacking the anterior tip in the CAMSM, but these lack most of their teeth, so these do not match Carter’s description either. However, a fragmentary rostrum identified as belonging to Carter’s collection (CAMSM B20659) possesses markedly curved teeth (Figs. 1 and 2). This is probably the material used by Carter to define the species (‘campylodon’ meaning ‘bent tooth’), as the mandibular teeth appear markedly recurved compared to the (pre)maxillary teeth, matching Carter’s description. Owen (1851: Pl. XXV) figured this specimen, which seemed to include a much longer portion of the rostrum at that time, thus possibly extending up to two feet. By all means, all articulated rostra discussed by Carter and the teeth he likely figured belong to the unreworked part of the West Melbury Marly Chalk Formation, not from the Late Albian of the Gault Formation. While the presumed holotype of Ichthyosaurus campylodon cannot be located with certainty, there is an abundant material from the Grey Chalk Subgroup in Carter’s Collection, and some specimens are clearly identified as being “syntypes”: CAMSM B20659 and a series of teeth CAMSM B20644_58, containing the ones likely figured by Carter (1846b) (Figs. 1 and 2, Fig. S1). This material can thus serve as a nucleus to redefine Ichthyosaurus campylodon, assess its supraspecific attribution and evaluate the diversity of the last European ichthyosaurs.

Systematic Palaeontology

ICHTHYOSAURIA Blainville, 1835	
THUNNOSAURIA Motani, 1999	
OPHTHALMOSAURIDAE Baur, 1987	
PLATYPTERYGIINAE Arkhangelsky, 2001 (sensu Fischer et al., 2012)	

PERVUSHOVISAURUSArkhangelsky, 1998

Type species–Pervushovisaurus bannovkensis (Arkhangelsky, 1998)

Additional included species–Pervushovisaurus campylodon (Carter, 1846a; Carter, 1846b) nov. comb.

Emended diagnosis (from Fischer et al., 2014a) —Platypterygiine ophthalmosaurid characterized by the following autapomorphies (those marked by an asterisk cannot be assessed in the material referred to Pervushovisaurus campylodon): presence of foramina along the ventral premaxillary–maxillary suture*; presence of a semi-oval foramen on the lateral surface of the premaxilla, anteroventral to the external naris*; presence of lateral ridges on the maxilla*; presence of wide supranarial ‘wing’ of the nasal (a similar structure, although much smaller, is present in ‘Platypterygius’ australis and Acamptonectes densus)* (see Kear, 2005; Fischer et al., 2012, respectively); robust splenial markedly protruding from the external surface of the mandible; root with quadrangular cross-section, with the cementum forming prominent 90° angles.

Pervushovisaurus is also characterized by the following unique combination of features: secondarily closed naris surrounded by foramina* (as in ‘Platypterygius’ sachicarum and ‘Platypterygius’ australis (see Paramo, 1997; Kear, 2005, respectively), and in Simbirskiasaurus birjukovi, although the ‘anterior’ naris is still present in this taxon (Maisch & Matzke, 2000; Fischer et al., 2014a)); elongated anterior process of the maxilla, reaching anteriorly the level of the nasal (unlike in Aegirosaurus leptospondylus, Sveltonectes insolitus and Muiscasaurus catheti) (Bardet & Fernández, 2000; Fischer et al., 2011a; Maxwell et al., 2015, respectively); rostrum straight (unlike in ‘Platypterygius’ americanus, ‘Platypterygius’ sachicarum, ‘Platypterygius’ australis and possibly Muiscasaurus catheti, where it is slightly curved anteroventrally Romer, 1968; Paramo, 1997; Kear, 2005; Maxwell et al., 2015, respectively); straight, non-recurved tooth crowns (unlike in Sveltonectes insolitus, Muiscasaurus catheti) (Fischer et al., 2011a; Maxwell et al., 2015, respectively).

Stratigraphic range–Early-middle Cenomanian, Late Cretaceous.

Geographic range–Europe–western Russia.

PERVUSHOVISAURUS CAMPYLODON (Carter, 1846a) nov. comb. Figs. 1–3

1846a Ichthyosaurus campylodon—Carter	
v 1846b Ichthyosaurus campylodon—Carter	
v 1851 Ichthyosaurus campylodon—Owen	
1922 Myopterygius campylodon—Huene	
v 1960 Myopterygius campylodon—Delair	
v 1972 Platypterygius campylodon—McGowan	
v 2003 Platypterygius campylodon—McGowan & Motani	

Syntype series and lectotype–CAMSM B20644, CAMSM B20646 to CAMSM B20658, a series of teeth (including a tooth likely figured in Carter, 1846b); CAMSM B20659, a partial rostrum, all from the West Melbury Marly Chalk Formation (Early Cenomanian), Cambridgeshire, UK. CAMSM B20659 is here formally designated as the lectotype (ICZN Articles 72.1.1, 73 and 74). Other specimens of the series (CAMSM B20644, CAMSM B20646 to CAMSM B20658) are thus designated as paralectotypes.

Referred specimens–CAMSM B20671a and CAMSM TN282, two partial rostra from the upper (chalky) part of the Cambridge Greensand Member (earliest Cenomanian), Cambridgeshire, UK (the specific locality of CAMSM B20671a is recorded: Barrington); NHMUK 33294 partim, a nearly complete tooth lacking the apex and the distal part of the root, from the West Melbury Marly Chalk Formation at Isleham, Cambridgeshire, UK; NHMUK R49, a series of articulated teeth from the West Melbury Marly Chalk Formation at Lydden Spout, Folkestone, UK.

Figure 3 Rostra referred to Pervushovisaurus campylodon (Carter, 1846a).

(A–D) CAMSM TN282, a partial rostrum possibly from a juvenile specimen. (A) Ventrolateral view. (B–C) Details of the teeth. (D) Detail of the premaxillary overbite. (E–F) CAMSM B20671a, a partial rostrum of a large specimen. (E) Lateral view. (F) detail of the mid-snout teeth. (G) Detail of the premaxillary overbite.

Emended diagnosis–Pervushovisaurus campylodon characterized by the following autapomorphy: slight overbite (3–4 cm). Pervushovisaurus campylodon is also characterized by the following unique combination of features: crown with rugose texture (shared with Aegirosaurus sp., ‘Platypterygius’ hercynicus and ‘Platypterygius’ sp. specimens from France and UK Fischer et al., 2011b; Fischer et al., 2014b; Fischer, 2012); acellular cementum ring of mid-snout teeth can possess shallow apicobasal ridges and furrows (shared with ‘Platypterygius’ australis) (Maxwell, Caldwell & Lamoureux, 2011).

Type horizon and locality–Lower Cenomanian of the Grey Chalk Subgroup, Upper Cretaceous. Cambridge area, Cambridgeshire, UK.

Remarks–The designated lectotype (CAMSM B20659), many teeth from rest of the syntypic series (CAMSM B20644, CAMSM B46_58) and the referred rostra (CAMSM B20671a, CAMSM TN282) each exhibit diagnostic features (Table 2). This material can be combined into a morphologically and spatiotemporally homogenous series that is distinguishable from the other ophthalmosaurid ichthyosaurs for which rostral and dental features have been reported.

Table 2 Distribution of the diagnostic features of Pervushovisaurus campylodon among the available specimens.

Osteological feature	Carter’s teeth (CAMSM B20644, CAMSM B46_58a)	Syntypic rostrum (CAMSM B20659)	Referred rostra (CAMSM B20671a, CAMSM TN282)	
Long maxilla	?	Y	Y	
Prominent root angles	Y	Y	Y	
Rugose enamel	Y	Y	Y	
Ridged acellular cementum ring	Y	Y	Y	
Thickened splenial	?	?	Y	
Straight rostrum	?	?	Y	
Overbite	?	?	Y	

CAMSM TN283, a large rostrum also originating from the Grey Chalk Subgroup, Cambridgeshire, closely resembles CAMSM B20671a and CAMSM TN282, but the autapomorphies of Pervushovisaurus campylodon cannot be evaluated unambiguously in this specimen; it is thus referred to as Platypterygiinae indet. A series of teeth and tooth bearing elements from the West Melbury Marly Chalk Formation collections of the NHMUK (NHMUK R1916, NHMUK R2335, NHMUK R2339, NHMUK 41895, NHMUK 47233, NHMUK 49911, NHMUK 52819) are, similarly, compatible with Pervushovisaurus campylodon in terms of tooth crown shape and size, and maxilla anterior extension but cannot be unambiguously referred to as Pervushovisaurus campylodon; these are thus referred to as Platypterygiinae indet. as well.

PLATYPTERYGIINAE indet

Referred specimen (see Table 1 for details)–CAMSM B20645; CAMSM TN283; CAMSM unnumbered; NHMUK R13; NHMUK R49; NHMUK R1916; NHMUK R2335; NHMUK R2339; NHMUK R2385; . NHMUK 5648; NHMUK 33294 partim; NHMUK 41367; NHMUK 41895, NHMUK 47233, NHMUK 49911, NHMUK 52819.

Note—CAMSM B20645 is a tooth is part of the type series of Ichthyosaurus campylodon, but lacks the diagnostic features of Pervushovisaurus and Pervushovisaurus campylodon; it is thus regarded as Platypterygiinae indet. OPHTHALMOSAURINAE Baur, 1887 sensu Fischer et al. (2012)	
OPHTHALMOSAURINAE indet.	

Referred specimen–Unknown specimen number, holotype of Ichthyosaurus kokeni (Broili, 1908).

Stratigraphy–Grodischter Schichten; middle and upper ‘Neocomian’.

Location–Vicinity of Hannover, Germany.

Synonymy

1908 Ichthyosaurus kokeni Broili: 432

Discussion—Ichthyosaurus kokeni is based on a basioccipital, and a partial humerus, a paddle element and several centra from the “Grodischter” schist, considered as Hauterivian in age by Broili (1908). A peripheral groove surrounds the condyle of the basioccipital, as in ophthalmosaurine ophthalmosaurids (Fischer et al., 2012). The humerus has three facets and the ulnar facet appears deflected posterolaterally with respect to the sagittal plane, further indicating ophthalmosaurine affinities (Fischer et al., 2012). The material lacks autapomorphies or unique combination of features and should therefore be regarded as a nomen dubium, assigned to Ophthalmosaurinae indet.

Description and Comparison of Pervushovi- saurus Campylodon

Premaxilla (CAMSM B20659; CAMSM B20671a; CAMSM TN282; Figs. 2 and 3)–The premaxilla is markedly elongated and has a semi-circular cross-section. Fossa praemaxillaris is a deep and continuous sulcus that is segmented anteriorly in a series of aligned foramina. As in Suevolevithan (Maisch, 2001) and some other Cretaceous ichthyosaurs (V Fischer, pers. obs. on unpublished material from the Albian of France, 2012), a complex network of the shallow grooves radiates from these foramina and textures the lateral surface of the premaxilla. In the anterior third of the rostrum, the dental groove is slightly constricted between functional teeth, forming subtle pseudo-alveoli. The labial wall of the dental groove then becomes straight and thickens posteriorly. The premaxilla forms a slight overbite (4–5 cm), a unique feature among ophthalmosaurids. This overbite is genuine because premaxillary and dentary teeth are still tightly interlocked in the anterior part of the rostrum in these specimens (CAMSM TN282, CAMSM B20671a).

Maxilla (CAMSM B20659; CAMSM B20671a; CAMSM TN282; Figs. 2 and 3)—The anterior process of the maxilla is elongated and its external extent reaches the level of emergence of the nasal, as in many platypterygiines, except Aegirosaurus, Sveltonectes, and Muiscasaurus (Romer, 1968; Kirton, 1983; Bardet & Fernández, 2000; Sirotti & Papazzoni, 2002; Fischer et al., 2011a; Fischer et al., 2011b; Maxwell et al., 2015) (note that Sirotti & Papazzoni (2002) interpreted the rostrum upside-down).

Dentary (CAMSM B20659; CAMSM B20671a; CAMSM TN282; Fig. 3)–The dentary is elongated, semi-circular and slightly deeper than the premaxilla. Fossa dentalis is narrow and ends anteriorly as a series of aligned foramina. Like in the premaxilla, the labial wall of the dental groove is constricted between functional teeth in the anterior third of the dentary. It is straight, unlike in some other platypterygiines (‘Platypterygius’ australis, ‘Platypterygius’ americanus and ‘Platypterygius’ sachicarum (Romer, 1968; Paramo, 1997; Kear, 2005)). The dentary is reduced anteriorly, creating an overbite.

Splenial (CAMSM B20671a; CAMSM TN282; Fig. 3)—The symphysis is 535 mm long in CAMSM TN282. The splenials are markedly thickened ventrally near the end of the symphysis, similar to the condition seen in Pervushovisaurus bannovkensis and regarded as one of the autapomorphies of this taxon (Fischer et al., 2014a).

Dentition (CAMSM B20644; CAMSM B20646_58; CAMSM B20659; CAMSM B20671a; CAMSM TN282; Figs. 1–3)—The crown is conical, robust, and covered by rugose enamel (as in Aegirosaurus sp., ‘Platypterygius’ hercynicus and Platypterygius sp. Fischer et al., 2011b; Fischer et al., 2014b; Fischer, 2012, respectively). Smaller specimens like CAMSM TN282 tend to have slenderer teeth. The acellular cementum ring is ridged on large teeth, but only apically, as in ‘Platypterygius’ australis (Maxwell, Caldwell & Lamoureux, 2011). The root possesses markedly flattened surfaces (mostly anterior and posterior ones); the root cement forms protruding ridges in between these facets, forming prominent and sharp ridges with a 90° angle cross-section, as in Pervushovisaurus bannovkensis (see Fischer et al., 2014a). This marks a sharp increase of the ‘diameter’ of the tooth, unlike in ‘Platypterygius’ hercynicus and many other isolated teeth from the Cambridge Greensand Member (Kuhn, 1946; Fischer et al., 2014b), where the diameter increases gradually. Numerous apicobasal ridges texture the labial and lingual surfaces of the root. Some of the dentary teeth of Carter’s syntype (CAMSM B20659) are markedly bent inwardly, which lead Carter to propose the name “campylodon” for reception of this material. However, slightly bent teeth are commonly encountered in many ichthyosaur specimens (Sollas, 1916; McGowan & Motani, 2003). While the dental grooves of the dentary appear indeed slightly oblique with respect to the sagittal plane, the strong bend appears here to result from diagenetic compression. I consider this feature as poorly diagnostic, and only very few isolated teeth exhibit a similar curvature of the root.

Cluster Dendrogram Results

The cluster dendrogram analysis resulted in a similar groupings than in Fischer et al. (2016). One exception is the displacement of ‘Platypterygius’ hercynicus and ‘Platypterygius’ americanus, two taxa with slightly smaller crowns, to the Generalist guild, from the Apex Predator guild (Fig. 4). These taxa remain clustered with ‘Platypterygius’ australis and Brachypterygius extremus within an Apex Predator guild if a 50% completeness threshold is applied to the raw data, however (see Fig. S3). Within the Apex Predator guild, Brachypterygius extremus, Pervushovisaurus bannovkensis and Pervushovisaurus campylodon form a cluster. Confidence values are slightly increased in the new version of the cluster dendrogram, with an average bootstrap of 0.148 (vs 0.122 in (Fischer et al., 2016)) and an average approximate unbiased P value of 0.989 (vs 0.982 in Fischer et al. (2016)).

Figure 4 Feeding ecology of the last ichthyosaurs.

(A) Cluster dendrogram resulting from the analysis of the ecomorphological dataset and showing separation of three main guilds. (B) Detail of spalled and subsequently polished apex in CAMSM TN283 (Platypterygiinae indet., closely resembling Pervushovisaurus campylodon).

Discussion

Generic attribution of large Albian-Cenomanian platypterygiines—The type material of Platypterygius platydactylus and Ichthyosaurus campylodon are barely overlapping, precluding an unambiguous referral to that genus. At the current state of knowledge, ‘Platypterygius’ australis and ‘Platypterygius’ campylodon do not share apomorphies; their rostral and dental similarities are plesiomorphic for platypterygiinae (Fischer et al., 2012). Most importantly, two peculiar features of Ichthyosaurus campylodon are shared with Pervushovisaurus bannovkensis: the prominent ridges forming 90° angles formed by the root cement in middle jaw/snout teeth and the ventrally protruding splenials. The type and only specimen Pervushovisaurus bannovkensis also exhibits a slight overbite (Fischer et al., 2014a), but the absence of teeth in situ precludes an unambiguous assessment of this feature in that taxon. Other differences between Pervushovisaurus bannovkensis and the syntypic material of Ichthyosaurus campylodon are the relatively smaller teeth in Pervushovisaurus bannovkensis, despite a seemingly larger skull size. The presence or absence of the other autapomorphic features of Pervushovisaurus bannovkensis cannot be assessed on material presently available of I. campylodon. Because of the similarities between Pervushovisaurus bannovkensis and I. campylodon, I propose to refer the species I. campylodon to the genus Pervushovisaurus. While additional specimens are certainly required to better assess whether Pervushovisaurus campylodon and Pervushovisaurus bannovkensis are conspecific or not, this is another important step in the clarification of Cretaceous ichthyosaur taxonomy.

Because Platypterygius as traditionally conceived is a wastebasket taxon, incorporating taxa distantly related to the Aptian type species Platypterygius platydactylus, assigning Cretaceous specimens to this genus, by default is not advisable (Fischer et al., 2016). However, the genus-group name Myopterygius Huene, 1922 is available. It was erected for a series of species: Ichthyosaurus campylodon, Ichthyosaurus strombecki (=nomen dubium Fischer et al., 2016), Ichthyosaurus hildesiensis (=nomen dubium Fischer et al., 2016), Ichthyosaurus kokeni (here regarded as Ophthalmosaurinae indet. see above), Ichthyosaurus indicus (=nomen dubium Fischer et al., 2016) and Ichthyosaurus marathonensis (=Ichthyosaurus australis (see Zammit, 2010)). There are thus two remaining candidates for the type species of Myopterygius: I. campylodon and I. marathonensis (= ‘Platypterygius’ australis). But there are no systematic rules regarding the designation of originally included nominal type species; the ICZN lists rules and best practices in Recommendations 69A.1–10.

On one hand, the species Ichthyosaurus campylodon is the first one on the list of species originally referred to Myopterygius. Before proposing the name Myopterygius, Huene (1922: 98) refers to the aforementioned species as the “Campylodongruppe” of Lydekker, reinforcing the idea that he probably intended Ichthyosaurus campylodon to be the equivalent of a type species for the genus Myopterygius. Resurrecting Myopterygius for reception of Ichthyosaurs campylodon would thus match the original interpretation of Huene, in a binomial that is still abundantly found in several museum collections across Europe. Such a move would match recommendations 69A.7, 69A.8, 69A.9, 69A.10 of the ICZN code, because I. marathonensis was poorly known when Huene published his work.

On the other hand, the species ‘Platypterygius’ australis is now known by abundant, excellently preserved material (Wade, 1984; Wade, 1990; Kear, 2005; Zammit, Norris & Kear, 2010) and could thus better positioned to fix an important genus-rank name (ICZN recommendation 69A.1). Currently, the number of specimens referred to as ‘Platypterygius’ campylodon is much larger than those referred to as ‘Platypterygius’ australis, but the novel features found in the syntypic series of ‘Platypterygius’ campylodon might result in a smaller number of specimens referable to this species.

Two additional factors need be considered here: the similarity between Ichthyosaurus campylodon and Pervushovisaurus bannovkensis, which indicate congeneric relationship between these two taxa and the unclear phylogenetic relationships among platypterygiine ichthyosaurs with subdivided nares (compare Fischer et al., 2014a; Fischer et al., 2016; Maxwell et al., 2015). This leaves two distinct solutions: (i) transfer I. campylodon and Pervushovisaurus bannovkensis to Myopterygius, with I. campylodon as the type species, and declare Pervushovisaurus as a junior synonym of Myopterygius. The status and generic attribution of ‘Platypterygius’ australis would be left undecided until a comprehensive study on the relationships of that taxon is undertaken, possibly needing a new genus-rank name for that taxon. (ii) Move Ichthyosaurus campylodon to Pervushovisaurus and leave the status of both Myopterygius and ‘Platypterygius’ australis undecided until further study on this later taxon. In order to move forward and stabilise the complex taxonomy of Cretaceous ichthyosaurs, I opt here of the second solution, which leaves the possibility to resurrect Myopterygius with ‘Platypterygius’ australis as its type species, but such a decision is beyond the scope of this paper.

The diversity of the last European ichthyosaurs—The results from the cluster dendrogram analysis refine the claim for the presence of diversified ichthyosaur ecomorphs during the Early/earliest Cenomanian, as ‘Platypterygius’ americanus carries the Generalist guild up to the Early Cenomanian, even though the end its biozone is poorly constrained. Pervushovisaurus campylodon and Pervushovisaurus bannovkensis, two of the last ichthyosaurs, are tightly clustered within the Apex predator guild. Numerous other ichthyosaur specimens are present in the Grey Chalk Subgroup collections of the CAMSM and NHMUK (excluding the Cambridge Greensand member). These remains—mainly isolated teeth, centra and some basicranial bones—are compatible with derived platypterygiines and resemble ‘Platypterygius’ hercynicus (Kuhn, 1946; Kolb & Sander, 2009; Fischer, 2012), although with a slightly larger tooth size, and the specimen of ‘Platypterygius’ cf. kiprijanoffi described by Bardet (1989) from the Cenomanian of northwestern France. I have been unable to find other specimens that unambiguously possessed the unique dental and rostral features of Pervushovisaurus in the CAMSM, NHMUK and RBINS collections. There are two non-mutually exclusive reasons for this: (i) the prominent root ridges might be restricted to a small region of the snout and (ii) two weakly divergent platypterygiine species might be present in the Grey Chalk Subgroup. This latter possibility is exemplified by NHMUK 41367, a partial rostrum that lacks an overbite (Fig. 5), thus differing from the material hereby assigned to Pervushovisaurus campylodon. Because the overbite in Pervushovisaurus campylodon appears more strongly expressed in the smallest rostrum (CAMSM TN282) than in the largest (CAMSM B20671a), it is possible that this feature vary with ontogeny. It is however unlikely that this feature completely vanish with adulthood, because specimen CAMSM B20671a, which belong to end of the spectrum of parvipelvian skull size, still possesses a noticeable overbite. Sexual dimorphism is also a possibility, but it cannot be tested with the material currently at hand.

Figure 5 Possible second taxon in the Grey Chalk Subgroup.

(A) Right lateral view. (B) Anterolateral view. Note the lack of a premaxillary overbite, as opposed to Pervushovisaurus campylodon, but the otherwise very similar teeth and rostrum shape, suggesting a similar ecological niche.

If present, any additional ichthyosaur species in the Grey Chalk Subgroup appear generally similar to Pervushovisaurus campylodon in terms of general tooth shape and inferred ecological niche. These taxa would fall within the ‘Apex predator’ niche, having absolutely large teeth and robust, relatively large, and heavily worn crowns (apex broken and polished). An example of intense wear can be seen on the rostrum CAMSM TN283 referred to Platypterygiinae indet. (Fig. 4B): one of the crowns has a significant portion of its apex spalled obliquely and polished. This is a rare wear stage for ichthyosaurs but common in so-called hypercarnivorous forms like the geosaurine metriorhynchid Dakosaurus maximus (Young et al., 2012) or tyrannosaurid theropods (Schubert & Ungar, 2005). Collectively, this suggests that Pervushovisaurus spp. and several coeval ichthyosaurs from the Cenomanian of western Europe occupied an apex predatory niche with a large body size, as indicated by isolated large centra and humeri in the CAMSM and NHMUK collections. The Cenomanian ichthyosaur record from the Grey Chalk Subgroup thus conforms to the global pattern of a two-step decline, ichthyosaurs being restricted to a single morphotype and ecological guild from the Early Cenomanian onwards: a large and long-snouted predator with robust teeth.

Yet, the small overbite in Pervushovisaurus campylodon raises questions regarding its function—if any. Moderate to large overbite evolved among leptonectid ichthyosaurs during the Early Jurassic (Huene, 1951; McGowan, 1986; McGowan, 1989; McGowan, 2003; Lomax, 2016). Overbite is not recorded in ichthyosaurs after the Toarcian; this feature thus re-evolved in Pervushovisaurus campylodon (or its ancestor if this feature is also present in Pervushovisaurus bannovkensis) after a 73 million years hiatus. A series of hypothetical functions of the sometimes extreme overbites seen in leptonectid ichthyosaurs have been made in the past (McGowan, 1979; Riess, 1986), including predatory (like a swordfish) and tactile (like a narwhal) functions (reviewed in Fischer, Guiomar & Godefroit, 2011). Leptonectids and Pervushovisaurus campylodon exhibit complex network of shallow grooves radiating from the anterior part of the fossa praemaxillaris, but such structure is also present in taxa with no overbite, such as Suevoleviathan (Maisch, 2001) and yet undescribed forms from France (V Fischer, pers. obs., 2012). These groove probably housed blood vessels, but their concentration in the rostral tip might suggest a sensory function, as in the recently described fossil phocoenid porpoise Semirostrum cerutti, which likely used its long dentary overbite to probe the sediment (Racicot et al., 2014). However, Pervushovisaurus campylodon clearly differ from the aforementioned taxa in having much stouter and larger rostrum and teeth and a much less conspicuous overbite, which might thus not yield any obvious functional advantage. Nevertheless, the presence of such a feature among Cretaceous ichthyosaurs illustrate the previously unappreciated phenotypic diversity of ichthyosaurs during this system.

Supplemental Information

Supplemental Information 1 Supplementary information and figures

Click here for additional data file.

Data S1 Ecomorphological dataset

Click here for additional data file.

I warmly thank Matt Riley (Sedgwick Museum, University of Cambridge, UK), Sandra Chapman and Paul Barrett (Natural History Museum, London, UK) for their care during my visits and Erin Maxwell (Staatliches Museum für Naturkunde , Stuttgart, Germany) for fruitful discussions on the taxonomy of Platypterygius and for providing data on American taxa. Finally, I would like to express my gratitude to Neil Kelley (Vanderbilt University, Nashville, TN, USA) and Marta Fernandez (Universidad Nacional de La Plata, La Plata, Argentina) for providing insightful and constructive comments, which substantially improved all aspects of this contribution.

Institutional abbreviations

CAMSM Sedgwick Museum of Earth Sciences, Cambridge University, Cambridge, UK

RBINS/IRSNB Royal Belgian Institute of Natural Sciences, Brussels, Belgium

NHMUK Natural History Museum, London, UK

Additional Information and Declarations

Competing Interests

Author Contributions

Data Availability

New Species Registration

The author declares there are no competing interests.

Valentin Fischer conceived and designed the experiments, performed the experiments, analyzed the data, contributed reagents/materials/analysis tools, wrote the paper, prepared figures and/or tables, reviewed drafts of the paper.

The following information was supplied regarding data availability:

The raw data has been supplied as Supplementary File.

The following information was supplied regarding the registration of a newly described species:

Publication LSID: urn:lsid:zoobank.org:pub:019DACEA-EBBE-4FAE-B885-3A9E5B1E1315.

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
