# Peer review of "Taxonomy of Platypterygius campylodon and the diversity of the last ichthyosaurs"

_PeerJ, doi:10.7717/peerj.2604_

## Round 0.1 · original submission · Minor Revisions

Reviewer 1 suggests that the guild analysis seems a bit out of place relative to the rest of the text, and is (in their opinion) a minor update of previous work that doesn't fit with the main aims of the paper. Given that the analysis isn't really addressed in the discussion and that P. campylodon (the focus of the paper) isn't included in the analysis, I agree. Could P. campylodon be added (at least some of the measurements seem to be preserved)? Or is it too incompletely known? Please consider strategies to better integrate the work into the manuscript.

Reviewer 1 has some concerns about broader taxonomic implications of moving 'P.' australis into Myopterygius. Please consider these and address them either in your response letter or in the revised manuscript. The reviewer presents some alternative actions that could be undertaken.

Both reviewers have concerns as to the usage of the terms "holotype" vs. "lectotype" for P. campylodon. In my reading of the situation (and in agreement with the reviewers), "lectotype" is the correct term, but I would suggest reference to and citation of the relevant ICZN articles in your manuscript to address this.

Note that both reviewers have included marked-up PDFs with additional comments.

I would present the supplementary ecological dataset in a different format--e.g., tab-delimited text file. In the current presentation, it would be difficult for others to rerun the analysis without alignment errors between the taxon list and measurement table.

It seems a shame to bury the Ophthalmosaurinae indet. reassessment in the supplementary information; the information will be lost to most readers. Please move it into the main body of the text (perhaps at the very end of the descriptions), and include a sentence noting how it is relevant to the overall issues you address in the discussion.
* * *
Minor notes: In abstract, correct spelling of "Icthyosauria" in the first sentence. Later in the abstract, the wording should be "A holotype" rather than "An holotype" (to conform with common usage).

·

Basic reporting

.

Experimental design

no comments

Validity of the findings

Results are well supported. Inferences and speculations are clearly identified as such. So I have no further comments

Additional comments

Most of my comments have been recorded directly on a PDF copy the draft manuscript. In brief summary, this manuscript is worthy to be published after minor revisions. It clarified an important issue (i.e. taxon delimitation of the youngest ichthyosaurs) and, on this basis, update the analysis of their diversity (both taxonomic and ecological). The article could be of interest not only to ichthyosaur specialist but to researchers working on marine vertebrates. Ichthyosaurs, sensu lato, are an iconic group of tetrapods secondary adapted marine life. The extinction of the group (not related to a catastrophic event) is still debated. Trying to understand this issue needs, necessary, a robust alpha-taxonomic basis. That is, a consensus as robust as possible on low level (= species level) taxon delimitation.
My mayor concern is related with type-designation criteria. I think that criteria established by the ICZN (Articles 72.1.1, 72.2 9 and 73) are not completely satisfied. The author designates the CAMSM B20659 as holotype of Pervushovisaurus campylodon nov.comb. (chosen among assumed Carter´s syntype series) . I am not convinced that this is correct as Fischer (present contribution) is not an original designation so this specimen must not be considered as holotype. Even if we accept that the series CAMSM B20644 - CAMSM B20659 collectively constitute the name-bearing type (=syntype series) of Ichthyosaurus campylodon Carter 1846, the CAMSM B20659 can be designated as lectotype. In any case, I don´t think that the CAMSM B20659 can be designated as holotype as Fischer (present contribution) is not the original author of Ichthyosaurus campylodon = Platypterygius campylodon = Pervushovisaurus campylodon. (see comments on the pdf file).
Minor comments on figures: as the taxomic history of names is a central issue of this contribution (particularly Carter´s type serie) maybe a figure of original labels (or photos of the material that permit to see these details) could help to follow the authors arguments exposed on “State of the art” section.
Finally, as I am not a native speaker, I don´t know if the English is unambiguous and/or if the English grammar and sentence structure are correct.

·

Basic reporting

I have made several minor corrections and comments throughout the text, in the attached PDF but otherwise the basic reporting appears good.

Experimental design

The primary aim of this paper is to clarify the history, morphology and taxonomy of 'Platypterygius' campylodon, a long-standing but relatively poorly studied Cretaceous ichthyosaur. This is a useful endeavor and the author has made valuable contributions by apparently relocating the original syntype series, and providing good figures and detailed comparisons with related species, as well as a detailed account of the confusing taxonomic history.

The paper also includes a cluster analysis of ecological guild structure of Cretaceous ichthyosaurs based on skull and tooth morphology. I find this part of the study somewhat out of place here for two reasons. 1) It appears at that 'P.' campylodon is not included in this analysis - despite being the focus of this paper. 2) This analysis essentially replicates a very recent publication by the author (Fischer et al. 2016, Nature Communications), with minor corrections and additions to the underlying dataset resulting in the movement of two taxa from the 'apex predator' to the 'generalist guild.' While I understand the desire to published a corrected analysis - it seems insufficiently integrated into the main aim of this paper. I would recommend dropping the guild analysis entirely unless it can be more clearly linked to the reanalysis of P. campylodon - and/or the novel results relative to the very recently published version can be better highlighted, again ideally within the context of the primary goals of the study.

Validity of the findings

Overall, I find the basic findings of the paper to be well reasoned and well supported. This work fits nicely in the arc of Dr. Fischer's work over recent years to clarify the relationships, morphology and evolution of Cretaceous ichthyosaurs. I do have a handful of questions and concerns, which are detailed in notes in the attached PDF but I will summarize here:

1) The author selects a 'holotype' from among the original type series. I believe this should more accurately be called and considered a 'lectotype' as it is (apparently) derived from the original syntype series described by Carter. If it is the case that this *specific specimen* from among those figured and described by Carter was intended as the holotype, that needs to be explained more clearly. Also, in that case, the 'syntypes' described here would actually become paratypes. Please review the nomenclature used in the paper and ensure that it is accurate and adheres to ICZN guidelines.

2) The author makes a pretty strong case for using the historically well-established genus 'Myopterygius' as a replacement for 'Platypterygius' for P. campylodon. He states:

(lines 375-377) "Resurrecting Myopterygius for reception of Ichthyosaurs campylodon would thus match the original interpretation of Huene, in a binomial that is still largely found in several museum collections across Europe."

and

(lines 384-386) "...Ichthyosaurus campylodon better represents the original intention of Huene and matches recommendations 69A.7, 69A.8, 69A.9, 69A.10 of the ICZN code, because I. marathonensis was poorly known when Huene published his work."

However, instead the author decides to transfer Myopterygius to 'P. australis' apparently on the grounds that that taxon is represented by abundant, well preserved and more complete material. Further justification for this transfer is given: "because Fischer et al. (2016) found that ‘Platypterygius’ australis is distantly related to Platypterygius platydactylus" however those results are not explicitly reproduced, or recounted here. Fischer et al. 2016 instead stated: "It is still premature to make a taxonomic decision on Platypterygius." It is difficult to see what major findings presented here would have changed this.

While I think technically legal, and not unreasonable, I believe the transfer of 'P.' australis to Myopterygius here is somewhat inadvisable as it has sweeping taxonomic implications for a well studied taxon that is not the focus of this work. The taxonomic history of P. australis is already complicated, and while transferring it to a new (or in this case, old) genus could be justified, I think that would be better achieved as part of a study that more comprehensively detailed the morphology and relationships of the species, or, even better, a detailed treatment of Platypterygius as a whole, building on the results of Fischer et al. 2016 and others.

It might be that one motivation for the resurrection of Myopterygius is to get the name 'out of the way' so that P. campylodon could be transferred to Perushovisaurus. However this does not seem absolutely necessary to me. I think P. campylodon could be referred to Perushovisaurus on the grounds given in this study (although see concerns about ontogenetic polarity of some characters in comments on pdf) without taking up the status of Myopterygius or 'P.' australis for the time being. That would be the approach I recommend here I think.

3) The final section of the paper speculates about the function of the overbite observed in P. campylodon, relative to more extensive (in some cases MUCH so) overbites in other ichthyosaur species, and other aquatic taxa. The overbite shown here seems quite modest (indeed it might be exaggerated on figure 1 -- see note), and possibly also related to allometric growth as it is weakly developed on smaller specimens. In this case I wonder whether it really invites specific functional considerations, certainly any kind of specialized sensory function seems pretty weakly indicated by morphology.

Additional comments

See comments on attached PDF. I think this is sound work that should certainly be published, I hope that my comments and suggestions are of use.

---

## Round 0.2 · Minor Revisions

Thank you for your thorough revision and careful attention to the comments from the reviewers. The paper, in my view, is nearly ready to go. The final item I would like to see changed is to combine the two .txt data analysis files into a single file, to avoid any confusion by readers. Once I have this change, I can accept the manuscript in very short order.

---

## Round 0.3 · accepted · Accept

Thank you for your quick attention to the last request for revision!